# Hemolysis Control in the Emergency Department by Interventional Blood Sampling

**DOI:** 10.3390/jpm13040651

**Published:** 2023-04-10

**Authors:** Hyeseung Lee, Heekyung Lee, Changsun Kim, Hyungoo Shin, Inhye Lee, Yihyun Kim

**Affiliations:** 1Department of Emergency Medicine, Guri Hospital, Hanyang University College of Medicine, Guri 11923, Republic of Korea; 2Department of Emergency Medicine, Yongin Severance Hospital, Yongin 16995, Republic of Korea; 3Department of Philosophy, University of Nevada, Las Vegas, NV 89154, USA

**Keywords:** blood sampling methods, hemolysis rate, intravenous catheter, venipuncture

## Abstract

The hemolysis rate in the emergency department (ED) is higher compared to that in other departments. We propose a new blood sampling technique without repeated venipuncture to reduce hemolysis and compare the hemolysis rate between blood collected by this method and that collected with an intravenous (IV) catheter. This prospective study included a nonconsecutive sample of patients visiting the ED (aged ≥ 18 years) of a tertiary urban university hospital. The intravenous catheterization was performed by three pre-trained nurses. The new blood collection technique involved sample collection without removing the catheter needle, performed immediately before the conventional method (through an IV catheter), without additional venipuncture. Two blood samples were collected from each patient using both the new and conventional methods, and the hemolysis index was evaluated. We compared the hemolysis rate between the two methods. From the 260 patients enrolled in this study, 147 (56.5%) were male, and the mean age was 58.3 years. The hemolysis rate of the new blood collection method was 1.9% (5/260), which was significantly lower than that of the conventional method (7.3%; 19/260) (*p* = 0.001). The new blood collection method can reduce the hemolysis rate as compared to the conventional blood collection method.

## 1. Introduction

During hemolysis, red blood cell (RBC) membranes rupture, resulting in the release of hemoglobin (Hb) into the serum. Inappropriate blood sample collection is a major cause of hemolysis and the primary cause of sample rejection, accounting for approximately 60% of rejected specimens [1,2,3].

The American Society of Clinical Pathology has defined a hemolysis rate of ≤2% as the benchmark for the best sample collection practice. However, several studies have reported a hemolysis rate of >2% in the emergency departments (EDs) of hospitals, and the hemolysis rate in EDs has been reported predominantly in comparison to that in other departments [2,4,5,6,7]. We hypothesize that the blood collection method in EDs may contribute to the high hemolysis rate. Blood samples are mainly collected using a straight stick needle in other departments (the standard method), while in EDs, they are mainly collected using peripheral intravenous (IV) catheters, wherein the line is initiated to increase efficiency and decrease patient discomfort by avoiding a second venipuncture (the conventional method) (Figure 1).

Drawing blood through IV catheters is associated with a significantly higher risk of hemolysis than via the conventional method using straight stick needles [8,9,10,11]. Nevertheless, blood sample collection via an IV catheter has been conventionally used and accepted as a standard nursing practice in EDs. This is because drawing blood with a straight stick needle requires repeated venipuncture, increasing patient discomfort and ED workload [8]. These disadvantages are more prominent in patients with difficult venipuncture access, who are admitted predominantly to the ED [8]. Moreover, repeated blood sampling with straight stick needles (the standard method) is difficult in EDs in general. Therefore, in this study, we proposed a new blood sampling technique to reduce hemolysis without repeated venipuncture and compared the hemolysis rate between the new blood collection method and the conventional method using an IV catheter.

## 2. Material and Methods

### 2.1. Study Design and Setting

This prospective study included a nonconsecutive sample of patients admitted to the ED from March 2020 to May 2020 in a tertiary urban university hospital, which has an average of approximately 50,000 visits per year. It was ethically approved by the Institutional Review Board of the Hanyang University Guri Hospital in September 2018 (Gyeonggi-do, Republic of Korea; 25 September 2018). Prior to the start of the experiment, informed consent was collected from both the enrolled patients and the practitioners (nurses and laboratory technicians).

### 2.2. Study Population

The participants of this study were patients (aged ≥ 18 years) admitted between 9:00 a.m. and 12:00 p.m. on working days, requiring blood sampling for laboratory tests. Patients with underlying diseases or conditions that affect hemolysis, including autoimmune hemolytic anemia, sickle cell disorder, thalassemia, hereditary spherocytosis, and pyruvate kinase deficiency, were excluded. Patients with hyperacute symptoms, including chest pain, dyspnea, focal neurologic deficits, and altered mental status, were also not included, as it is difficult to obtain informed consent from these patients in an emergency setting.

### 2.3. Study Protocol and Data Collection

According to our experimental protocol, three nurses with >5 years of clinical experience in the ED performed the blood collections. Prior to the study, the nurses completed a retraining course that included the new blood sampling method implemented in this study. They placed an IV catheter (BD Angiocath PlusTM; Becton Dickinson Infusion Therapy Systems Inc., Sandy, UT, USA) in an upper or lower limb vein and collected blood according to the following protocol: (1) applied a tourniquet and inserted a catheter into the patient’s upper or lower limb; (2) in the case of flashback, removed the catheter lever lock plug; (3) connected a syringe and removed the tourniquet; (4) drew blood using a stylet needle (specimen 1); (5) threaded the catheter and withdrew the stylet needle; (6) connected a new syringe and drew blood again (specimen 2); (7) collected each specimen in a new vacutainer and rotated; (8) labelled the vacutainer with a randomly preassigned number; (9) transferred the vacutainer to the core laboratory immediately after collection using a pneumatic tube; (10) contacted the laboratory technician directly (Figure 2).

In accordance with the protocol, the laboratory technician collected the two vacutainers as soon as they were contacted and centrifuged them according to the manufacturer’s instructions. The Hb level in the samples was also quantified as the hemolysis index (HI) using the chemistry analyzer AU5800 (Beckman Coulter, Brea, CA, USA). Based on previous studies, the association between the HI value and the concentration of free Hb was considered as follows: HI value of 0, Hb < 50 mg/dL; HI value of 1, Hb < 100 mg/dL; HI value of 2, Hb < 200 mg/dL; HI value of 3, Hb < 300 mg/dL; HI value of 4, Hb < 400 mg/dL; HI value of 5, Hb > 500 mg/dL [12,13]. The laboratory technicians were blinded to the study specimens and phlebotomy techniques. They only recorded the patients’ identification numbers, assessed the time and HI values, and randomly assigned the number of each vacutainer (specimen).

The data, including patient information (sex, age, underlying diseases, blood pressure, and presenting symptoms), were recorded in a predesigned form. The location of the IV catheterization (upper arm, antecubital, forearm, hand, or leg), catheter size (18–24 gauge), and catheterization difficulty (1 = very easy; 2 = easy; 3 = neither easy nor difficult; 4 = difficult; 5 = very difficult) were also recorded. All of the data were collected daily by the researcher CK.

### 2.4. Main Outcome

Blood samples with >100 mg/dL of free Hb can cause non-specific binding in laboratory tests. Therefore, serological testing is not recommended for these serum samples [14]. By the biochemical protocol of our institution, detailed serologic testing is not performed for samples with an HI of grade 2 (≥100 mg/dL of Hb) or above. Therefore, hemolysis was defined as an HI of grade 2 or higher. The presence of hemolysis was compared between the two methods.

### 2.5. Statistical Analyses

The continuous variables were presented as means and standard deviations (SDs), and the categorical variables were presented as frequencies and percentages (%). The statistical analysis included the Wilcoxon test for the paired samples (for the continuous variables) and the McNemar test for the paired samples (for the categorical data). The statistical analyses were conducted using SPSS (version 27.0 for Windows, Chicago, IL, USA). The statistical significance was set at a *p* value of <0.05.

## 3. Results

### 3.1. Demographics and Baseline Characteristics

Overall, 260 patients were enrolled in this study (Figure 3). Of these, 56.5% were male, and the mean age was 58.3 ± 20.3 (SD) years. Of the total patients enrolled, 19.6%, 16.5%, and 7.3% of the patients had pre-existing hypertension, diabetes mellitus, and renal insufficiency, respectively. The presenting symptoms included abdominal pain (68.1%), dizziness (18.8%), headache (8.1%), and others (5.0%).

The IV catheterization details (IV catheterization location, size, and difficulty) are presented in Table 1. Approximately three-quarters (72.3%) of the patients underwent catheterization in the antecubital fossa (40.0%) and forearm (32.3%). A large-gauge IV catheter (18 gauge) was used in 84.6% of the patients. The nurses reported that most (97.3%) of the IV catheterizations were not difficult.

### 3.2. Main Outcome

A comparison of the HI between the new and conventional blood collection methods is presented in Table 2. The hemolysis rate of the new blood collection method was 1.9%, which was significantly lower than that of the conventional method via an IV catheter (7.3%) (*p* = 0.001). The mean HI value of the new blood collection method was 0.16 ± 0.026, which was significantly lower than that of the conventional method via the IV catheter (0.36 ± 0.046) (*p* < 0.001).

## 4. Discussion

The new blood collection method proposed in this study reduced the hemolysis rate (by 1.9%) compared to the conventional blood collection method using an IV catheter (7.3%) (*p* = 0.003). Similar to our results, previous studies have reported an ED hemolysis rate higher (6–30%) than the 2% benchmark established by the American Society for Clinical Pathology [2,4,5,6,7]. Although the reasons for the higher hemolysis rates in EDs are likely multifactorial, blood collection using an IV catheter has been proposed to greatly contribute to them. Johnson & Johnson explained the cause of the high hemolysis rates with the use of straight stick needles. As catheter material development technologies advance, softer plastic has been used in catheters to reduce irritation to the inner walls of blood vessels. Softer catheters can be used well under positive pressure conditions for the administration of fluids or drugs, but they can collapse under negative pressure conditions, including blood collection, resulting in turbulence and hemolysis. Thus, hemolysis caused by using an IV catheter is greater than that caused by using a straight stick needle, which provides an intact inner lumen unchanged by negative pressure [5]. The new blood collection method proposed here can also provide an intact inner lumen during negative pressure conditions of blood sampling without the need for a second needle insertion, thereby lowering the hemolysis rate by reducing turbulence.

Hemolysis is the breakdown of RBCs, which can affect laboratory results. Blood samples with >100 mg/dL of hemoglobin can cause non-specific binding in serologic tests, causing sample rejection and resulting in unnecessary repeated testing. It causes additional patient discomfort, higher costs, increased ED workload, delays in definitive acute care, and an increased ED stay. These delays severely affect patients with time-sensitive diseases, including myocardial infarction and acute stroke [2,15]. Although blood samples with HI values of 1 (Hb concentration: 50–100 mf/dL) are not rejected by laboratories, they can also produce unreliable test results. Perovic et al. reported that hemolysis significantly affected the values of α-amylase, alkaline phosphatase, aspartate aminotransferase, bilirubin, creatine kinase (CK), CK-MB, gamma-glutamyl transferase, iron, lactate dehydrogenase, magnesium, potassium, total protein, and uric acid, even at HI = 1 [16]. Moreover, Bais et al. reported that contemporary troponin-I and high-sensitivity troponin-T tests were sufficiently affected at relatively low degrees of hemolysis based on their tests [15]. Thus, although reduction of the rejection rate is important, reduction of the HI value itself (the degree of hemolysis) is also important. The median HI value of the new method was 0.16 ± 0.026 in this study, which was significantly lower than that of the conventional method (0.36 ± 0.049).

A peripheral IV catheter, 20 gauge or larger, placed in the antecubital fossa or forearm is recommended for contrast computed tomography (CT), because a flow rate of 3 mL/s or higher is necessary for injecting contrast media [17]. Contrast CT scans are widely used during ED management [18,19,20]; thus, a 20-gauge or larger IV catheter is usually preferred in EDs. At our institution, an 18-gauge catheter has been primarily used for IV lines, and it was most frequently used (84.6%) during the study period. Small-sized catheters (22 or 24 gauge) were also inserted in some patients with thin or fragile blood vessels (15.4%).

The internal diameter of the straight stick needles used in this study was smaller than that of the outside catheter. The 18-gauge catheter usually has a 20-gauge inside needle. A small catheter diameter may cause higher hemolysis. A small internal diameter can result in a large negative force being applied to the blood, inducing shear stress on the RBCs during blood sampling [21,22]. Thus, the smaller diameter of the inside straight needle than the outside catheter may offset the effect of the straight needle on reducing hemolysis. However, this phenomenon (size-dependent effect) was not observed when a sufficiently large gauge needle (18–22 gauge) was used [22]. Thus, most of the results with an 18-gauge catheter may not be significantly affected by the difference in size between the outer catheter (18 gauge) and the inner needle (20 gauge). However, the results obtained with the 22- and 24-gauge catheters could be affected. Thus, the results of this study may be difficult to apply to the small-gauge catheter (22 gauge or smaller) used during the blood sampling. The sample size for blood sampling using small-gauge catheters was small in this study; thus, further research is needed.

In laboratories, it is common practice to detect and report hemolysis using visual inspection of the specimens after centrifugation and comparing them with the hemolytic chart. However, visual assessment is subjective, and several studies have reported little agreement between the actual hemoglobin levels and the hemolysis grades [23,24]. In this study, the degree of hemolysis was determined using the HI for consistency between the laboratory technicians; this method has been proven to be accurate for identifying the level of hemolysis [13,25].

This study had some limitations. Inexperienced nurses may pull the syringe strongly or have long tourniquet times, which can affect the degree of hemolysis [5]. In this study, only three experienced nurses were involved in the catheterization practice to reduce the inexperience bias. Nevertheless, bias due to differences in practitioners cannot be excluded.

Most patients (95%) included in this study complained of abdominal pain (68.1%), dizziness (18.8%), and headache (8.1%). As all types of patients visiting the ED were not included in the study, the results of this study may not be generalizable to all ED situations.

## 5. Conclusions

The new blood collection method can reduce the hemolysis rate as compared to the conventional blood collection method using an IV catheter.

## Figures and Tables

**Figure 1 jpm-13-00651-f001:**
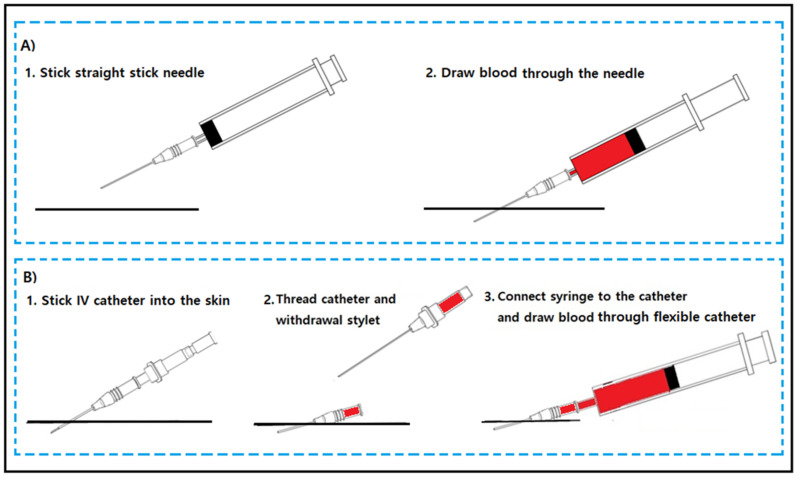
The standard blood collection method (**A**) and the conventional blood collection method in EDs (**B**).

**Figure 2 jpm-13-00651-f002:**
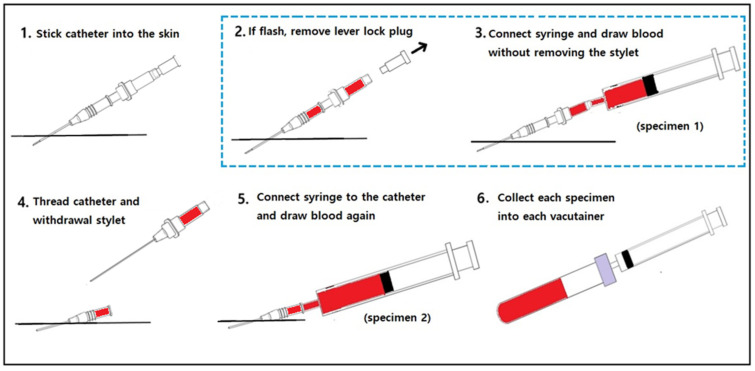
The blood sampling protocol in this study: the new blood collection method (2, 3) and the conventional blood sampling (4, 5) method.

**Figure 3 jpm-13-00651-f003:**
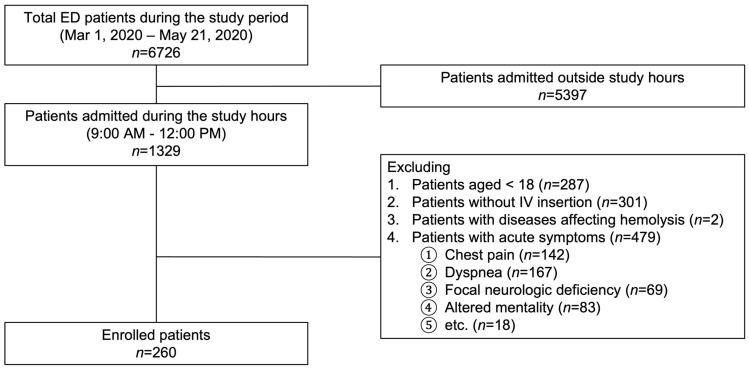
Flowchart of the study. ED, emergency department; IV, intravenous.

**Table 1 jpm-13-00651-t001:** Baseline characteristics of the study participants.

Variables	Value
Age, year, mean (SD)	59.3 (20.3)
Sex, male, *n* (%)	147 (56.5)
Underlying diseases, *n* (%)	
hypertension	51 (19.6)
DM	43 (16.2)
renal insufficiency	19 (7.3)
Presenting symptoms, *n* (%)	
abdominal pain	177 (68.1)
dizziness	49 (18.8)
headache	21 (8.1)
etc.	13 (5.0)
Blood pressure, mmHg, mean (SD)	
systolic	136 (30.7)
diastolic	79 (16.3)
Location of IV catheterization, *n* (%)	
Upper arm	2 (0.8)
antecubital	104 (40.0)
forearm	84 (32.3)
hand	65 (25.0)
leg	5 (1.9)
Catheter size, *n* (%)	
18 gauge	220 (84.6)
22 gauge	18 (6.9)
24 gauge	22 (8.5)
Difficulty of Catheterization, *n* (%)	
very easy	185 (71.2)
easy	39 (15.0)
neither easy nor difficult	29 (11.2)
difficult	6 (2.3)
very difficult	1 (0.4)

IV, intravenous; SD, standard deviation.

**Table 2 jpm-13-00651-t002:** Comparison of the hemolysis index between the new blood collection method and the conventional blood collection method.

	Blood Collection Method		
	New Method	Conventional Method Using IV Catheter	*p* Value
Hemolysis Index (HI)			
0	224 (86.1)	199 (76.5)	
1	31 (11.9)	42 (16.2)	
2	3 (1.2)	7 (2.7)	
3	2 (0.8)	9 (3.5)	
4	0	3 (1.2)	
5	0	0	
Non-hemolysis vs. hemolysis			
non-hemolysis (HI ≤ 1)	255 (98.1)	241 (92.7)	0.001
hemolysis (HI ≥ 2)	5 (1.9)	19 (7.3)	
Mean of HI (SD)	0.16 (0.026)	0.36 (0.049)	<0.001

## Data Availability

The datasets used and/or analyzed during the current study are available from the corresponding author on reasonable request.

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
