# Peer review of "Hemolysis Control in the Emergency Department by Interventional Blood Sampling"

_jpm, 2023, doi:10.3390/jpm13040651_

Round 1
Reviewer 1 Report
The manuscript seems to be valuable for proposing a new blood collection method. I have minor comments.
1. (Abstract) It seems that the conventional method was conducted always after the new collection method. Did the order of the collection methods affect the results? In addition, was the conventional method conducted even if hemolysis occurred in the new collection method? Moreover, weren’t there any patients who dropped out after the new collection method was used?
2. (Abstract) What statistical analysis method was used for calculating p-value?
3. (Methods) If paired-samples were used for the analysis, it is better to use McNemar Test instead of chi-squared test.
4. (Methods or results) How about presenting a flowchart of selecting participants for the study? Were data of all the enrolled participants obtained in this study?
Author Response
To: Reviewer #1
Thank you for your detailed and rigorous comments. We further revised our manuscript based on all of your comments as follows. Sincerely.
Best regards.

Reviewer 2 Report
Manuscript jpm-2337513 entitled “A New Blood Collection Method to Reduce Hemolysis in the Emergency Department”. Please notice the following:
General view: The manuscript presented a novel and good practical idea using good English and grammar. A little degree of simplification is a must to make the section more readable for the common reader and higher understanding and readability. The manuscript could be accepted for publication after minor revision.
Title: Clear to a greater extent but preferred to be modified into “Hemolysis Control in the Emergency Department by Interventional Blood Sampling”.
Abstract: Clear, concise, and informative.
Keywords: Clear but please rearrange the keywords in alphabetical order.
Introduction: Comprehensible, informative, and clear
The aim: Clear, well-defined, and informative.
Materials and methods: Clear, systematic, informative, and brief.
Results: Do not list many numbers in the results section and you can use the percentages only to avoid redundancy in the results section. Other than that, the results are novel, clear, and informative.
Discussion: Consider the listing of percentages only instead of using numbers and percentages. Other than that, the discussion section is clear, informative, contributes to knowledge, and is comprehensible with a good level of comparison and speculation.
Conclusion: Clear, concise, and informative.
Authors’ contributions: Not listed.
Acknowledgment: Clear and informative.
Funding: Clear and informative.
References: Sufficient as only 32% (8 out of 25) were published in the past five years.
Tables: Well organized and presented.
Figures: Well organized and presented.
Author Response
Reviewer #2
To. Reviewer #2
Thank you for your detailed and rigorous comments. We further revised our manuscript based on all of your comments as follows. Sincerely.
Best regards.
